# Polarization of Macrophages in Human Adipose Tissue is Related to the Fatty Acid Spectrum in Membrane Phospholipids

**DOI:** 10.3390/nu12010008

**Published:** 2019-12-18

**Authors:** Rudolf Poledne, Hana Malinska, Hana Kubatova, Jiri Fronek, Filip Thieme, Sona Kauerova, Ivana Kralova Lesna

**Affiliations:** 1Centre for Experimental Medicine, Institute for Clinical and Experimental Medicine, 140-21 Prague, Czech Republic; hana.malinska@ikem.cz (H.M.); hana.kubatova@ikem.cz (H.K.);; 2Transplant Surgery Dept., Institute for Clinical and Experimental Medicine, 140-21 Prague, Czech Republic; jiri.fronek@ikem.cz (J.F.); filip.thieme@ikem.cz (F.T.); 3Anaestesiology, Resuscitation and Intensive Care Unit, Military University Hospital, 140-21 Prague, Czech Republic

**Keywords:** macrophages, membrane, inflammation, nutrition, omega-3 fatty acids

## Abstract

Residential macrophages in adipose tissue play a pivotal role in the development of inflammation not only within this tissue, but also affect the proinflammatory status of the whole body. Data on human adipose tissue inflammation and the role of macrophages are rather scarce. We previously documented that the proportion of proinflammatory macrophages in human adipose tissue correlates closely with non-HDL cholesterol concentrations. We hypothesized that this is due to the identical influence of diet on both parameters and decided to analyze the fatty acid spectrum in cell membrane phospholipids of the same individuals as a parameter of the diet consumed. Proinflammatory and anti-inflammatory macrophages were isolated from human adipose tissue (*n* = 43) and determined by flow cytometry as CD14+CD16+CD36^high^ and CD14+CD16−CD163+, respectively. The spectrum of fatty acids in phospholipids in the cell membranes of specimens of the same adipose tissue was analyzed, and the proportion of proinflammatory macrophage increased with the proportions of palmitic and palmitoleic acids. Contrariwise, these macrophages decreased with increasing alpha-linolenic acid, total n-3 fatty acids, n-3/n-6 ratio, and eicosatetraenoic acid. A mirror picture was documented for the proportion of anti-inflammatory macrophages. The dietary score, obtained using a food frequency questionnaire, documented a positive relation to proinflammatory macrophages in individuals who consumed predominantly vegetable fat and fish, and individuals who consumed diets based on animal fat without fish and nut consumption. he present data support our hypothesis that macrophage polarization in human visceral adipose tissue is related to fatty acid metabolism, cell membrane composition, and diet consumed. It is suggested that fatty acid metabolism might participate also in inflammation and the risk of developing cardiovascular disease.

## 1. Introduction

Despite large-scale and successful treatment of two major risk factors of atherosclerosis, high blood pressure and high LDL cholesterol concentration, cardiovascular disease (CVD) remains the leading cause of death. This is most likely due to the continuous increase in the prevalence of obesity worldwide [1,2] and, at the same time, to visceral ectopic fat enlargement associated with increasing proinflammatory status. It has been repeatedly shown that residential macrophages in adipose tissue play a pivotal role in the development of inflammation not only within this tissue [3], but also influence the proinflammatory status of the whole body. Monocytes released from the spleen [4] to blood circulation are partly caught in adipose tissue, and local inflammation leading to increased immune cell migration may accelerate this process. The presence of macrophages within adipose tissue in animal experimental models has been shown to be stimulated by diets high in fat and cholesterol, and the proportion of macrophages in some of these models represented almost 40% of all adipose tissue cells [5]. It has been repeatedly proven that adipose tissue inflammation is associated with atherosclerosis progression [6]. There is a widely recognized theory that this effect is mostly indirect due to the production of adhesive and proinflammatory cytokines by residential macrophages, as well as adipocytes [7].

Catching of macrophages within adipose tissue is related to the most important risk factors of atherosclerosis and CVD. Several molecular mechanisms of the role of macrophages during the long-term process of atherogenesis have been elucidated [8]. We recently showed that the proportion of metabolically normally stimulated macrophages in human adipose tissue is positively related to body mass index (BMI) [9], age (differently in men and women) [10], and is very closely positively related to non-high-density lipoprotein HDL) cholesterol concentration [11]. On the other hand, the properties and behavior of normally stimulated proinflammatory macrophages in adipose tissue (referred to as M1) are not yet fully understood. The common cause of the very close correlation of proatherogenic lipoprotein concentrations to polarization of adipose tissue macrophages [11] might be due to the mutual effect of diet on both parameters. Next, fatty acids (FAs) in phospholipids of adipose tissue cell membranes (probably the best parameter to document long-term saturated and unsaturated FA consumption) were analyzed and correlated to the proportion of proinflammatory adipose tissue macrophages. It can also be speculated that, similar to red cells whereby insulin sensitivity [12] and inflammation [13] are directly affected by changes in the cell membrane and its FA spectrum, the role in macrophages polarization could also be influenced. Consequently, the composition of cell membrane phospholipids within adipose tissue might be related to changes in the intracellular metabolism of macrophages and their polarization.

## 2. Materials and Methods

### 2.1. Living Kidney Donors

A total of 43 individuals (enrolled between July 2014 and June 2016) were fully informed about the process of kidney donation and transplantation and about adipose tissue sampling during organ cleansing before transplantation. All individuals signed informed consent forms and were interviewed about their medical history, diet, and major cardiovascular risk factors. This project was approved by the local Ethics Committee of the Institute for Clinical and Experimental Medicine and the Faculty Thomayer Hospital on 27 June 2012 under code number 1041/12 according to the Helsinki declaration of 1975 as revised in 2000, and the study was conducted in accordance with the approved protocol.

### 2.2. Tissue Samples

Samples of visceral adipose tissue (approximately 2 g) were obtained intraoperatively after hand-assisted laparoscopic nephrectomy to be immediately cooled and transferred to a laboratory. A part of these samples was used for analysis of the macrophage phenotype. Once visible blood vessels and connective tissue were removed, each sample was dissected (approximately 2 mm^2^). After shaking incubation of tissue samples with collagenase (2 mg/L) for 15 min (37 °C), the homogenate was filtered (50 μm) and centrifuged. The stromal vascular fraction (SVF) was purified twice by resuspension. The final SVF samples were analyzed immediately by flow cytometry (CyAn, Beckman Coulter, Brea, CA, USA). Only samples with viability greater than 75% were considered (measured using 7-aminoactinomycin D). Monoclonal antibodies and fluorochromes (CD14, Phycoerythrin-Cyanine, CD16, Phycoerythrin-Texas Red X (ECD), CD36, fluorescein isothiocyanate (FITC), and CD163 Phycoerythrin, PE/Clone RM 3/1) were used to define different subsets of monocytes/macrophages. The flow cytometry data were analyzed using Kaluza software (Beckman Coulter, Brea, CA, USA).

The method was described in detail earlier [10] with definition of two phenotypes. Based on data from the literature [13,14,15,16] and our recent results [17], we proposed that macrophages with a combined phenotype characterized by the expression of CD14 and CD16 and high phagocytic receptor CD36 expression [18] correspond to normally stimulated M1 proinflammatory macrophages calling them proinflammatory adipose tissue macrophages (ATMs). On the other hand, macrophages with no CD16 expression, but with CD163 positivity, could be considered M2 [19] and were referred to as anti-inflammatory ATMs. We are well-aware that this classification could oversimplify an in vivo situation in which a full phenotypic spectrum of transient phenotypes between M1 and M2 may exist and that this situation needs to be considered. These transition fractions represent 15 ± 2.3% of the total of macrophages within the adipose tissue of living kidney donors (LKDs) varying between 6% and 35%.

The other part of adipose tissue samples was stored at −80 °C and used for fatty acid (FA) analysis.

### 2.3. Fatty Acid Composition

Once all adipose tissue samples were collected and subsequently stored at −80 °C, they were removed from the freezer and slowly defrosted. The extraction, separation, and methylation of adipose tissue phospholipids were performed as previously described [18]. Total lipids were extracted with dichlormethane:methanol using a modified Folch method and phospholipids were isolated by thin-layer chromatography using hexane-diethyl ether-acetic acid (80:20:3, *v*/*v*) as the solvent system. The FAs in phospholipids were converted to methyl esters using a 1% solution of Na in methanol. The methyl esters were eluted with hexane and separated by gas chromatography using a Hewlett-Packard GC system with hydrogen as the carrying gas, a flame ionization detector, and a carbowax-fused silica capillary column (Varian, Palo Alto, CA, USA) [19]. Individual peaks of FA methyl esters were identified by comparing retention times with those of authentic standards (mix of standard FAs, Restek Corporation, PA, USA). The proportions of FAs (a spectrum of the 18 main FAs of interest) are given as the relative percentage of the sum of FAs analyzed. The relationships of eight FAs to the macrophage phenotype were considered. The other minor FA fractions did not display any significant effect in our analysis.

### 2.4. Biochemistry

Total cholesterol, triglyceride, and HDL-cholesterol fractions were determined from fasting blood samples obtained immediately before surgery (prior to anesthesia) using an enzymatic method (Hoffmann-LaRoche, Basel, Switzerland). C-reactive protein (hsCRP) was measured by immunoturbidimetric assay using a Cobas Mira Plus autoanalyzer (Hoffmann-LaRoche, Basel, Switzerland).

### 2.5. Subjects Diet Stratification

Our simple dietary frequency questionnaire was composed of only eight questions was completed and analyzed in 39 LKDs. The diet was analyzed by a food frequency questionnaire (in Appendix A), with special attention paid to the consumption of vegetable or animal fat for cooking and spreading, and fish and nut consumption. The highest score was obtained in individuals who consumed butter and lard for cooking, as well as for spreading (including cream/processed cheeses) on bread, but did not declare fish and nuts consumption. The lowest score was attained by individuals who declared only vegetable fat sources and very frequent consumption of fish and nuts.

### 2.6. Statistical Methods

Our data are presented as means with standard deviations for continuous variables. Categorial variables are expressed by the number of subjects. The parametric unpaired *t*-test could be used for continuous endpoints between LKDs and representative control because they passed the D′Agostino-Pearson normality test. The correlations of the normally distributed macrophage subpopulations and FA spectrum were documented by the coefficient of determination (*r*^2OK^), which was calculated with the Pearson method, including the *p*-value. All tests were two-tailed and the level of significance was set at 0.05. Statistical analyses were performed using Prism 6 (GraphPad Software, Inc., La Jolla, CA, USA).

## 3. Results

### 3.1. Study Subjects

All FA spectrum analyses were completed in 43 LKDs with a majority of maternal donors. A comparison of the characteristics of LKDs to the data of our sex- and age-matched controls selected from a representative Czech population sample [1] showed that both groups were fairly similar in all of the parameters followed (Table 1).

The LKDs were slightly leaner, with lower total cholesterol levels and a similar HDL/total cholesterol ratio (26.5% for LKDs and 27.1% for controls). There were no diabetics, as diabetes mellitus is an exclusion criterion for kidney donation. The relevant population was included to document that a relatively small LKD group does not represent any selection of “more healthy” individuals. Instead, the data are valid for the general population.

### 3.2. FA and ATM Relations

The proportion of proinflammatory ATMs in the group of healthy LKDs varied from 20% to more than 60%. Palmitate, as well as its desaturation product, i.e., palmitoleate, correlated positively with proinflammatory ATMs. The relation of palmitoleate was strongest among all of FA data (Figure 1).

Surprisingly, an opposite, but also significant, correlation was found for stearate. Because of the opposite trend of the two most important saturated FAs on macrophage polarization, a mutual effect of all saturated FAs was not found (data no presented). No relation was found for total monounsaturated FAs (MUFAs) despite the maximum positive relation for palmitoleate itself. A negative correlation of alpha-linolenic acid (ALA) with the proportion of proinflammatory ATMs was documented, and similar significant relations were evident for total n-3 polyunsaturated FA (PUFA) and the n-3/n-6 FA ratio. As there was no relationship of n-6 FA to the proportions of proinflammatory ATMs, only the effect of n-3 FA was considered. A highly significant negative relation of the proportion of eicosatetraenoic acid (EPA) to proinflammatory ATMs was shown (Figure 1). The strength of the relation remained unchanged when performing these correlations with the exclusion of men (*n* = 10).

A mirror picture was seen for the anti-inflammatory ATMs in adipose tissue, which varied from 20% to 70% (Figure 2). No trend in the proportion of palmitate to anti-inflammatory ATMs was observed, while stearic acid was significantly positively related to anti-inflammatory ATMs. It is of interest that the effect of palmitoleic acid on the proportion of both pro- and anti-inflammatory macrophages reached the highest significance. It is also interesting that of all the MUFAs measured, this relation was found only for palmitoleate, hence there was no effect on total MUFAs. Our data did not show any relation to the proportion of ALA to anti-inflammatory ATMs. A positive correlation was demonstrated for n-3 PUFAs and the n-3/n-6 ratio to anti-inflammatory macrophages.

To sum up, all changes in the FA spectrum in adipose tissue phospholipids to both the pro- and anti-inflammatory ATM phenotypes (Figure 1 and Figure 2) were interconnected.

### 3.3. Dietary Score and Proinflammatory ATM Proportion

The ratio of the dietary score to the proportion of proinflammatory ATMs and the dietary score are presented in Figure 3. The highest score, i.e., 16, was obtained in individuals who consumed butter and lard for cooking, as well as for spreading on bread, with no fish and nut consumption. The lowest score, i.e., 4, represented individuals with only vegetable fat in their diet and very frequent consumption of fish and nuts. The dietary score of LKDs was significantly related to the proportion of proinflammatory ATMs (*p* < 0.01, Figure 3). The same opposite relation was also documented for anti-inflammatory ATMs (Appendix A).

## 4. Discussion

For over a decade, it has been known that inflammation, especially the behavior of macrophages within adipose tissue, plays a key role in the risk of developing CVD. We documented that the proportion of proinflammatory ATMs is related to the three major risk factors of atherosclerosis, i.e., age (differently in men and women), increased BMI, and non-HDL cholesterol concentration [8,9,10]. In this study, we analyzed ATM polarization in relation to the composition of cell membrane phospholipids of human visceral adipose tissue together with the current diet of LKDs.

The polarization of proinflammatory ATMs in the group of healthy individuals was positively related to the proportion of palmitate (Figure 1), which is in agreement with its effect in some experimental models [20,21]. Surprisingly, the effect of stearate was the opposite. Although some findings have described certain differences between the metabolism of the most common saturated FAs (SAFAs) [22], it is difficult to explain these differences based on the published data, as well as our data. As the above-mentioned two most important SAFAs exerted opposite effects, the overall effect of SAFAs was not found.

Fatty acid synthesis in macrophages has been shown to accelerate M1 polarization in tissue culture [20]. De novo synthetized FAs are incorporated into plasma membrane phospholipids and might possibly directly influence macrophage polarization [20]. The proportion of palmitate as the final product of FA synthesis might therefore cause an increase in the proportion of proinflammatory ATMs in LKDs with enhanced FA synthesis in adipose tissue. This increasing effect of macrophage polarization was also documented in experimental models with stimulated lipogenesis [23]. In principle, the different effects of palmitate and stearate on the proportion of proinflammatory ATMs might be partly explained by compartmentalization of the complex proinflammatory effects within the cell [24]. Palmitate also stimulates atherogenesis in LDL KO knock out mice, affecting reverse cholesterol transport [25], as well as increasing the production of the proinflammatory cytokines interleukin (IL)-13, monocyte chemoattractant protein-1, and IL-6.

The most prominent correlation with proinflammatory ATMs was found in palmitoleate despite its minority within cell membrane phospholipids. The interest in this molecule has increased recently [24]. Although dietary palmitoleate intake is very low [26], it represented as much as 1–3% of the proportion of FAs in visceral adipose tissue phospholipids of our LKDs. Palmitoleate is described as a “lipokine” with multiple regulatory roles [27]. The anti-inflammatory effect of palmitoleate was repeatedly documented in several experimental models [27,28,29,30], as was in the stimulation of insulin sensitivity (for review, see [27]). However, the effect on insulin sensitivity has not yet been confirmed in a human study [31]. All the beneficial effects of palmitoleate do not correspond to its stimulation of proinflammatory polarization in human adipose tissue in our data. This discrepancy could be explained by the substantially different effect of cis-and trans-palmitoleate reported recently [32].

Contrary to palmitate and palmitoleate, n-3 FAs had an opposite effect on macrophage polarization. An increased proportion of ALA (Figure 1) was connected with a decreased proportion of proinflammatory ATMs, with a similar effect found for total n-3 FAs and the n-3/n-6 FA ratio. This is essentially consistent with tissue culture data of the influence of n-3 PUFAs [33]. It is well-known that dietary supplemented n-3 FAs are incorporated into the plasma membrane [34] and this enrichment of the cell membrane decreases the proinflammatory status in experimental models [35], whereas n-6 FAs are not effective [36,37].

The relation of increased EPA to decreasing proportion of proinflammatory ATMs was highly significant (Figure 1). This anti-inflammatory effect of EPA associated with increased fish oil intake was documented in circulatory monocytes [38]. Although this short-term experiment may not have affected adipose tissue macrophages, previous data are in agreement with our results. The increasing proportion of EPA in the cell membrane of adipose tissue is combined with decreasing polarization of proinflammatory macrophages and is essentially consistent with this FA effect in recent studies of sepsis treatment [35,39].

The proportion of proinflammatory ATMs is positively related to those of palmitic and palmitoleic acids and is a potentially proatherogenic phenomenon. A similar phagocytic phenotype was also identified in circulating monocytes [40] as CD14 and CD16 positive cells with the presence of the CD36 receptor. In 438 patients with chronic kidney disease, the expression of this molecule increased gradually with a decrease in glomerular filtration rate and predicted cardiovascular events. The fact that macrophages with this phenotype decreases with increasing proportions of ALA, n-3 FA, and EPA in our data is indirect proof of their protective role.

The effect of the proportion of FAs in cell membrane phospholipids on anti-inflammatory alternatively stimulated ATMs displayed a mirror pattern to proinflammatory ATMs, but the statistical power of the correlations was lower (Figure 2) (with the exception of a highly significant effect of palmitoleate). It should be stressed that the pro- and anti-inflammatory phenotypes do not total 100%. There are also intermediary macrophages representing, on average 15%, of total macrophages, varying between 6% and 37%. This variation is probably the reason for the relatively vaguer correlation of the different FA groups to the proportion of anti-inflammatory ATMs. The strong negative relationship of palmitoleate to the proportion of anti-inflammatory ATMs supports the unusual importance of this FA in macrophage polarization in human visceral adipose tissue.

Although the positive effect of n-3 FAs on triglyceridemia, hypertension, coagulation, and arrhythmia was summarized previously [41], the association of ALA intake with coronary heart disease was definitively proved only recently [42] in a meta-analysis of 14 prospective quantitative studies. The documented effects of essential FAs (specifically ALA) correspond to the analyzed diets of our subjects. The dietary score exhibited a significant effect on the proportion of proinflammatory ATMs (Figure 3). Although our questionnaire was rather simple and focused mainly on fish and nut consumption, the significant relation of the dietary score to the proinflammatory ATM proportion supports the possibility of a direct influence of these nutrients on macrophage polarization.

A healthy diet with the recommended increase in n-3 FA is able to shift polarization of macrophages in adipose tissue to a lower proportion of proinflammatory ATMs in adipose tissue. This applies mainly to the proportion of ALA of individual study subjects. The most significant effect was found for EPA present in fish and nuts. Thus, the importance of diet on changing the FA spectrum in the adipose tissue cell membrane was documented despite the relatively small number of individuals studied (limited number of LKD transplant procedures performed at the Institution within one year).

Comparing the effects of individual FAs on inflammation with our results is rather difficult, as the currently available data were obtained mainly from in vitro experiments. For example, the action of oleic acid itself was found to be anti-inflammatory (analyzed by the presence of the CD206 receptor) with an inhibitory effect of palmitate [43]. Palmitate has been shown to increase IL-13 production in macrophages [44], whereas n-3 PUFAs are stimulators of anti-inflammatory molecule (resorpin and protractin) production [45]. Although the effects of individual FAs in cell membrane phospholipids of human adipose tissue are more difficult to explain, their pro- and anti-inflammatory activities are, in principle, mostly in agreement with these in vitro data.

The main limitation of our study is that the FA spectrum was analyzed in phospholipids of the cell membrane of total adipose tissue and not separately in adipocytes and macrophages. However, when comparing the amount of phospholipids in the total of adipose tissue and its SVF fraction in six individual tissue samples, SVF (including numerous different cells and macrophages) represented less than 20% of total phospholipids of adipose tissue (with minor FA proportion differences between total adipose tissue and SVF, see Appendix A), so the FA spectrum represents an adipose tissue milieu and not the effect of local macrophages.

Proinflammatory polarization of ATMs by palmitate and palmitoleate and the opposite effect of ALA, n-3 FAs, n-3/n-6 FA ratio, and EPA potentially represent a new effect on the risk of developing CVD. The importance of our data obtained in human adipose tissue of healthy individuals is amplified by a similar effect of these FAs on the proinflammatory status of the whole body as measured by TNF-α concentration (data not shown). It should be stressed that, in addition to intracellular FA metabolism, a direct effect of diet was also documented.

## 5. Conclusions

Admittedly, the direct causality of the effect of FAs on inflammatory changes could be questioned, and an opposite effect of inflammation on the FA spectrum should be accepted. However, our data documented the effects of diet on macrophage polarization, making our conclusion more plausible. There is a potential synergy of dietary saturated FAs, which increase the LDL-cholesterol concentration and the proportion of proinflammatory ATM on the one hand, and an LDL-cholesterol-decreasing effect of n-3 polygenic FA and decreasing the proportion of proinflammatory ATMs on the other. This synergistic effect also relates to the diet consumed and might play a very important role in lifestyle changes aimed at reducing the risk of developing cardiovascular disease.

## Figures and Tables

**Figure 1 nutrients-12-00008-f001:**
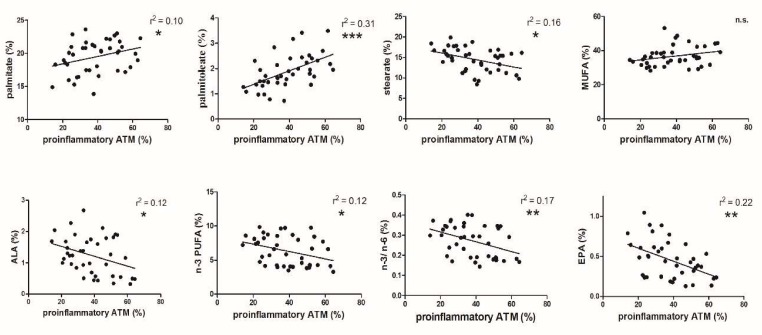
Correlation of the proportions of proinflammatory adipose tissue macrophages (ATMs) to different fatty acids (FAs) in phospholipids of adipose tissue cell membranes (n.s.: non significant, * *p* < 0.05; ** *p* < 0.01; *** *p* < 0.001). alpha-linolenic acid (ALA), n-3 polyunsaturated fatty acids (n-3 FUFA), n-3 polyunsaturated fatty acids/n-6 polyunsaturated fatty acids (n-3/n-6), eicosatetraenoic acid (EPA).

**Figure 2 nutrients-12-00008-f002:**
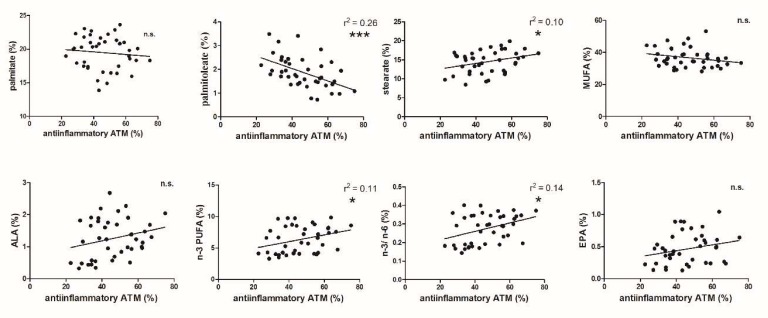
Correlation of the proportions of anti-inflammatory ATMs to different FAs in phospholipids of adipose tissue cell membranes (* *p* < 0.05; *** *p* < 0.001).

**Figure 3 nutrients-12-00008-f003:**
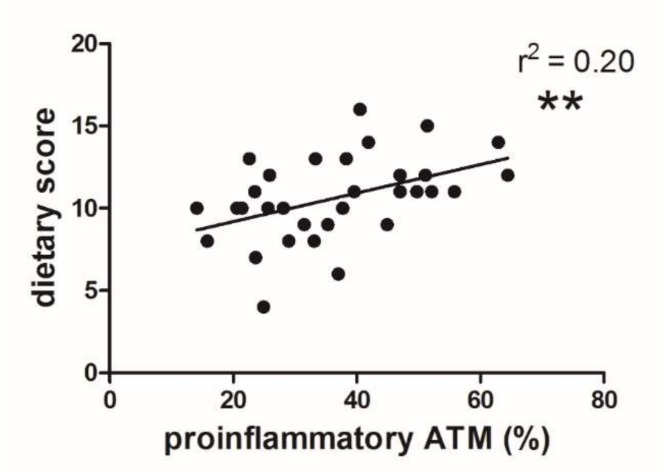
Correlation of the proportion proinflammatory ATMs to the dietary score (*n* = 39; ** *p* < 0.01).

**Table 1 nutrients-12-00008-t001:** Characteristics of the group of living kidney donors consisting of 27 women and 16 men and age and sex-matched controls from the 1% representative Czech population sample [1].

	LKDs	Controls
Age (years)	46.30 ± 9.87	45.77 ± 9.73
BMI (kg/m^2^)	25.95 ± 3.77	27.45 ± 6.03
Cholesterol (mmol/L)	4.40 ± 0.94	5.43 ± 1.09 ***
HDL-cholesterol (mmol/L)	1.16 ± 0.38	1.47 ± 0.34 ***
non-HDL-cholesterol (mmol/L)	3.24 ± 0.91	3.96 ± 1.08
Triglycerides (mmol/L)	1.51 ± 0.82	1.77 ± 1.21
hsCRP (mg/L)	1.24 ± 1.79	1.86 ± 2.36
Hypertension (Yes/No)	8/35	16/27
Diabetes mellitus (Yes/No)	0/43	3/40

Results are expressed as means ± standard deviation (SD), body mass index (BMI), high-density lipoprotein, HDL, C-reactive protein (hsCRP), living kidney donors (LKDs), *** *p* < 0.001.

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
