# Peer review of "Polarization of Macrophages in Human Adipose Tissue is Related to the Fatty Acid Spectrum in Membrane Phospholipids"

_nutrients, 2019, doi:10.3390/nu12010008_

Round 1

Reviewer 1 Report

R Poledne and colleagues have submitted an original article for publication to Nutrients. The subject is of interest but the work remains descriptive and has several gaps in data analysis. 

The study has a control population which is not use in data interpretation. WHat is it relevance for the work?

In the method section, the physical characteristics of the column use for FA profiling are missing.

The stratification of subjects according to food recording is unclear and this very simple questionnaire provides interesting data, but it is not sure that it is enough for correlation analysis with measured continuous variables. The test use for the correlation analysis is not mentioned.

In the description of the results, no correlation coefficients or R2 are provided, these parameters are of importance to characterize the intensity of the associations.

A major point to consider is the possibl egender effect in the correlations. It is well known that omega 3 metabolism differs between men and women. Gender should be considered as a covariable.

par 3.3: The last sentence needs to be explained: what is the link with non-HDL cholesterol here??

Discussion: line 272: THe authors could not clearly state this conclusion, although only 20% of the cells ara macrophages, we cannot exclude that a strong effect in macrophages PL has a significant impact at the level of the whole tissue (the proportion of proinflammatory macrophages varies from 20 to 80%.

minor points,

line 148: alpha linolenic acid 

Reviewer 2 Report

The section in the introduction on macrophage recruitment and inflammation of adipose tissue is unstructured, and the relationship between the components is not always obvious. You need to describe in more detail the mechanisms of the following processes:
“[12], [13]" What statistical criterion was used to determine the normal distribution? What method was used to calculate correlations? Figure 1. You need to add the exact r2 values ​​for each figure.

Reviewer 3 Report

In this this descriptive manuscript, Poledne et al elucidate a relationship between the polarization of macrophages in human adipose tissue with the fatty acid and phospholipid composition of the adipose tissue.  They find that the number of pro-inflammatory macrophages increases with palmitic and palmitoleic acid, and decreases with increased alpha-linolenic, omega 3 fatty acids, and eicosonpentanoic acid.  Pro-inflammatory macrophage was also correlated with dietary consumption of vegetable/fish vs. animal fat consumption.  This study was purely correlational, and the conclusions generally were supported by the data, however, numerous methodological considerations were noted.

Major concerns:

Numerous studies have shown that adipocytes from males and females have differential lipid metabolism (as reviewed in, Journal of Obesity & Metabolic Syndrome 2017;26:172-180), thus, the data should be separated by sex, with male and female subjects analyzed separately.

The authors sate that “Correlation analysis was performed using biostatistical GraphPad Prism software”.  This is not sufficient enough detail to understand the statistics used.  The authors should provide a detailed description of the stats, as well as a correlation coefficient for all of the correlations.

The rationale for Table 1 is not clear, since all of the samples from LKDs and controls were both used in all of the correlational analysis.

Minor concenrs:

Although differences in total omega 3 FAs is observed, there is also a difference in omega 3 to omega 6 FA ratio that appears to have a greater correlation coefficient.   The authors should report if there was a correlation with total omega 6 FA.

Reviewer 4 Report

In the manuscript by Poledne et al .the authors show an interesting correlation between macrophage polarization and fatty acid composition in adipose tissue of kidney donors. However, there are several concerns which should be addressed.

Major concerns:

1). The abstract needs to be re-written. As it is now, it has no flow. Starting from background information on macrophages in one sentence, it jumps to rather a long description of the methodology, lacking a sentence on the main aim of the work. Why it is important to study FA composition in relation to macrophages?

2) What is the proportion of pro-inflammatory and anti-inflammatory ATMs on the individual level? This should be stated as the range 20-60% or 20-70% in the group is too wide and non-informative. Consequently, how the FA composition is correlated with macrophage types on an individual level? This could have been shown by somehow marking the points in figures 1 and 2 (for example numbering them by subjects in open circles).

3) line 88 -89 the sentence is not clear, “varying between 6% and 35%” is this variation between the individuals? Then what is 15% and why it has so small deviation?

In connection to this, by which means the transition fraction was considered in your results?

5) Figure 3- the dietary score is correlated only with pro-inflammatory ATM, it would be good to show also data on anti-inflammatory ATM, is there a negative correlation with their proportion?

4) instead of writing “data not shown”, in lines 272 and 278 these data could be included in supplementary materials, which will be more informative than the questionnaire.

Minor comments

Figures 1 and 2 the axis titles are not visible, the font can be increased (the graph sizes could overall be increased for better visibility).

Line 214- “of proinflammatory” the word macrophage or ATM is missing

Line 228 –“polarization of phagocytic macrophages” -to be consistent with the rest of the text use proinflammatory instead of phagocytic.

Line 271 -SVF definition should be given.

Round 2

Reviewer 1 Report

The authors have corrected their manuscript in agreement with the comments.

Line 212 214: two sentences for the same observation

Author Response

The sentence was excluded.

Reviewer 3 Report

The authors have adequately addresssed my concerns.

Author Response

Thank you.

Reviewer 4 Report

The authors attempted to address the majority of the comments; however, some points still remain to be improved.

1). Particularly, the abstract did not become better. To my test, it is a bit flat and still lacks the information on why the authors decided to focus on FA composition? this sentence should be in the beginning, after the added “Data on human adipose tissue inflammation and the role of macrophages are rather scarce”, and should contain background information. Otherwise, it is really not clear how the hypothesis is generated.

It also contains too many experimental/technical details that make the reading difficult and can be removed from abstract, I think.

2). Abut figure 3 – “The same opposite relation was also documented for antiinflammatory ATMs (Figure not presented).”

If this is documented why to not present then? The suggestion is to include it as figure 3B

Author Response

The abstract was corrected according to your suggestions. The relationship of antiinflammatory macrophages to dietary score was included to the Supplementary material as Fig. 1